# Smart City for Sustainable Development: Applied Processes from SUMP to MaaS at European Level

**Francesco Russo and Corrado Rindone \***

Dipartimento di Ingegneria dell'Informazione, delle Infrastrutture e dell'Energia Sostenibile, Università degli Studi Mediterranea di Reggio Calabria, 89122 Reggio Calabria, Italy
**\*** Correspondence: corrado.rindone@unirc.it; Tel.: +39-3396641087

**Abstract:** Urban areas constitute one of the main issues of sustainability as defined by the United Nations with the Sustainable Development Goals (SDGs). The recent smart city concept represents a way for achieving the urban sustainability goals. The European Commission (EC) bases the smart city concept on three pillars: energy, transport and Information and Communication Technologies (ICT). The main objective of the paper is to investigate the European smart city process, by focusing on urban mobility and their interconnections with the other two pillars. The methodological approach of territorial planning is used by identifying the plan dimensions and then analyzing the processes at master and sectorial level. The applied processes are verified with a review of the European documents that constitute the rules for defining and implementing the smart city concept. European guidelines indicate the SUMP as the integrated master plan that contributes to reach the convergence among energy, transport and ICT processes. By focusing on people mobility sector, European cities are implementing the Mobility as a Service (MaaS) plan at the sectorial level. This implies the necessity to enhance the knowledge of mobility phenomenon, in relation to emerging ICT and their impact on energy consumptions. The contribution of the work is given by the identification of a planning and implementation path focused on smart city, in urban areas, which connects the general goals of Agenda 2030 with the daily implications for citizens and therefore with the specific results. The paper results are useful: from one side, for researchers that work on advancements of theories, and from another side, for planners and decision makers to explore the European attempts towards urban sustainability and the real implementations on urban mobility systems.

**Keywords:** smart city; planning; sustainable mobility; urban mobility; SUMP; MaaS; energy; ICT

## 1. Introduction

Cities are very complex systems due to the concentration of economic activities and social and environmental interactions. For these reasons, cities play a relevant role in the pursuing sustainability goals in the classical components related to social, economic and environmental development. The United Nations estimates that, by 2050, 66% of the world's population will live in cities, and this implies that urban challenges will strongly increase. Then, if the urban challenges are not faced with a correct approach, the sustainability goals cannot be achieved [1]. The United Nations confront this challenge with Agenda 2030, adopted in 2015, to increase "peace and prosperity for people and the planet, now and into the future" [2]. The agenda sets out 17 Sustainable Development Goals (SDG) defined by 189 targets and 261 indicators [3] that each state measures by means of national variables (Figure 1).

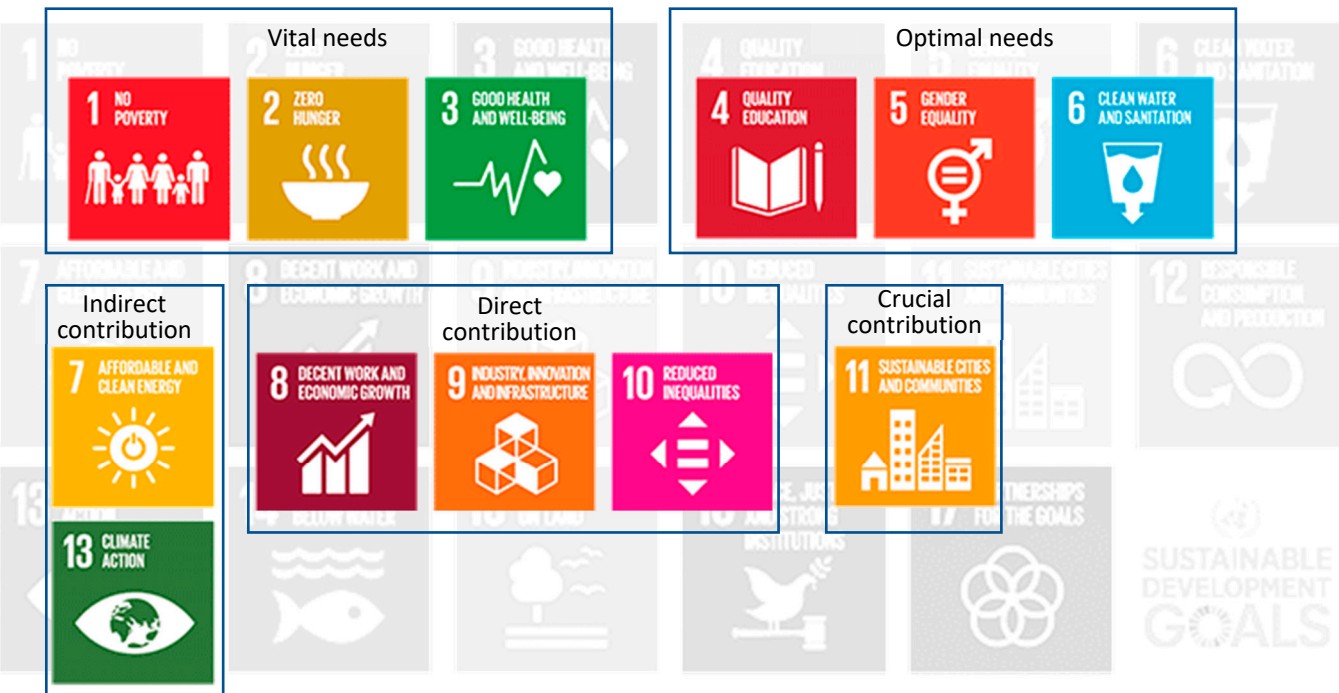

**Figure 1.** SDGs related to urban mobility (source: elaboration from UN, 2015 [2]).

Within the 17 SDGs, it is possible to identify two homogeneous classes that regard specific dimensions of sustainability [4,5]:

- "Vital Needs" that includes SDGs 1, 2 and 3, referring to basic needs of society to allow for survival (poverty, hunger, health);
- "Optimal Needs" that includes SDGs 4, 5 and 6, referring to the needs of education, equity and natural resources for allowing for growth.

The smart city concept represents a way to achieve the urban sustainability goals even if the connection between smart and sustainability cities is not obvious [6–8]. The concept is gaining relevance in research for developing urban theories, in politics for developing urban rules and in real practices for developing urban implementations [9]. The dynamic among theories, rules and implementations represents a further complexity, and the smart city contributes to a positive evolution of this dynamic [9,10]. For this reason, advancements in the single process and their application require more insights [11].

Urban transport is connected to different sustainable development goals. Challenges of mobility of people and goods are related to the difficulty of reaching a trade-off between "the right to access and mobility on the one hand, and the right to clean air and living quality on the other hand" in line with the dilemma described in the "Tragedy of the Commons" [12]. "Individual mobility rather belongs to the private right, a clean and safe environment shall be regarded as a commons to be safeguarded". In the urban sustainability perspective, the limitation of natural resources represents one of the main challenges in the current and future contexts and it generates the dilemma. For instance, the imbalance between mobility needs and limitation of energy resources is very challenging. Looking for a trade-off to solve this imbalance implies the need to study mobility from different perspectives and point of views.

Agenda 2030 dedicates specific attention to urban mobility within specific SDGs, targets and relative indicators. Figure 1 underlines the groups of SDGs and those that can have a positive contribution for sustainable urban mobility. It is possible to individuate the SGDs that provide the following ([5,13]):

- A crucial contribution to sustainable mobility: SDG 11 oriented to sustainable cities and communities;

- A direct contribution to sustainable mobility: SDG 8 oriented to economic growth; SDG 9 oriented to sustainable industrial development and resilient infrastructure; SDG 10 oriented to social sustainability;
- An indirect contribution to sustainable mobility: SDG 7 oriented to the use of renewable energy; SDG13 oriented to climate challenges.

This paper proposes an analysis of the advancements on the smart city process at European level, by focusing on the urban mobility development directions determined by the SDGs. The European Commission has promoted the smart city concept since 2012 by defining a set of rules in relation to three pillars: energy production and use, transport and mobility and Information and Communication Technologies (ICT) [14]. The emerging ICT have potentialities to find solutions to the urban sustainability. However, the isolated use of ICT is not the final solution. Data and information produced by ICT enhance the capabilities of Transport System Models (TSM) to reproduce the mobility phenomenon as a dynamic and evolving system [14]. At the same time, the combination among ICT, transport and energy empowers transport planning, supporting decision makers in configuring infrastructures and services that respond to mobility needs and, at the same time, limiting the use of natural resources and, in particular, energy consumptions [15].

One of the main examples of this development direction is represented by the Mobility as a Service (MaaS) paradigm [16]. MaaS is a user-centered form of mobility enabled by the combination of ICT and Transport System Models (TSM) potentialities linked to energy decisions. The final goal is to offers alternatives to the unsustainable mobility of people and goods to optimize the use of available resources, including energy ([4,17,18]).

In the face of the indications of the UN and the EU, the directions of sustainable development have not been pursued with strength and determination by the cities, as also highlighted in the 2020 report of the European Court of Auditors (ECA) [19], which reads: "there is no clear indication that cities are functionally changing their approaches. In particular, there is no clear trend towards more sustainable modes of transport". On this basis, it is possible to see that the smart city, in terms of theory rules and implementation at different planning time scales, is a direction that allows for the achievement of the sustainability objectives of Agenda 2030.

In the framework defined by Agenda 2030, schematized in Figure 1, the main goal of the paper is to identify the smart city as a central direction for achieving the crucial, direct and indirect goals identified by Agenda 2030 for urban areas.

The goal of the paper is pursued through three objectives. The first objective concerns the analysis of the processes applied at European level to implement the smart city, with reference to the interaction of theories, rules and implementations, highlighting the planning dimensions within which the smart city process develops and linking this new process to the most important territorial development processes of the last 50 years. The second objective concerns the methods of implementation of the smart city, in Europe, in the strategic horizon, defined as the master level. It is a question of verifying how it should be specified in terms of rules within the Urban Plans for Sustainable Mobility for people and freight. The third objective is even more specific and concerns how to implement the smart city in sectoral plans in short time horizons. It is a question of verifying how the development direction of the smart city becomes a reality usable by citizens in mobility through the MaaS.

Identifying the general goal and pursuing the three defined objectives is the novelty of the paper. In fact, there are no works in the literature that make it possible to directly link the focal goals, defined by the 2030 Agenda, with the daily life of citizens. The paper produces these successive zooms: starting from Agenda 2030, passing through the internationally consolidated planning tools, specifying what happens in the strategic masterplans and detailing with the operational tools. Reconstructing this path is particularly important because it allows us to verify to what extent countries and, subordinately, cities pursue sustainable development.

The contribution that the paper can give to scientific development is to identify the smart city as the main path to follow in mobility planning tools to pursue sustainability objectives.

The paper, to build this path, cites and connects various works present in the literature, but none of the known works defines the smart city direction as one of the most important directions to follow in the development of mobility plans at various time scales.

The results presented in this work can therefore be condensed into the identification of a homogeneous scientific and technical path that directly connects the general goals of Agenda 2030 with the results that can be evaluated ex ante, and therefore with the verification of the correct pursuit of the goals. The homogeneous path passes through an adequate territorial master plan to be developed in the SUMP (Sustainable Urban Mobility Plan) context, and an adequate sectoral plan for mobility of passengers to be developed in the MaaS context. The results are important because this sequence of plans and programs has not been studied in the literature, and it is therefore not available to technicians and researchers. Without having a clear and well-defined path available, the goals of Agenda 2030 remain only declarations of principle.

After this Introduction, the document consists of four sections. Section 2 first presents the territorial development process in the dynamics of its components (theories, rules, implementation) and then in the development of the different plans at different levels, presenting the main sectors of the smart city. Section 3 analyses how the smart city is applied at the European level in the master plan, focusing on specified theories, rules and implementations for sustainable urban mobility planning. Section 4 presents the application of the smart city in the urban mobility sectoral plans with the MaaS, verifying that, from a correct application, it is possible to arrive at the knowledge of the environmental impacts and, therefore, closing the circle, to know the indicators envisaged by Agenda 2030. The final section reports some conclusions regarding the different processes applied.

## 2. The Methodological Approach to Territorial Planning Process

### 2.1. Main and Specific Processes

The study of territorial processes implies the advancement of knowledge through a continuous process of interaction between two basic phases: theory and implementation. Territorial processes, unlike other real processes, must consider the rules that public decision makers give. These rules sometimes reinforce theoretical components, and other times the realizations have already taken place in reality [20]. It is therefore necessary to consider the dynamic process of successive advances between theory, rules and implementations.

The increasing knowledge related to progress in science and technology, from one side, and the increasing complexity, from another side, imply the need to reach a subsequent state of equilibrium among the three phases.

Cities comprehend a set of complex systems that interact in order to respond to the needs of people and businesses [21]. The increasing complexity of urbanization processes implies the necessity to reach a subsequent equilibrium among the three phases, which, in turn, are composed of specific processes: city planning theories are the object of urban studies, city rules are produced by urban policy makers and city development or real implementations are realized by economic operators. The smart city represents a possibility to reach an advanced equilibrium for achieving urban sustainability goals [9]. The new state of the main process can be reached following two circular ways: following the first way, the urban theories are transposed by decision makers into city rules that address real implementations; following the second way, city development influences city rules from which city planning theory emerges (Figure 2). Each positive exchange between two generic processes contributes to the new state and then to the increasing level of urban sustainability. The series of points that the urban system reaches are specific themes of discussion. In general, in the territorial processes, they need to follow the processes that move from one state to another one in a dynamic way, where each point constitutes a final step of a process and an initial point for another one.

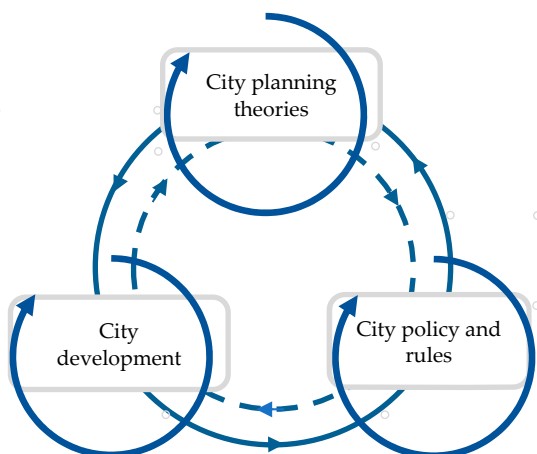

**Figure 2.** Main territorial planning processes.

*2.2. Planning Development Dimensions*

Historically, cities have been involved in a variety of economic, social and environmental processes.

Land uses, and therefore the allocation of activities, are relevant historical processes.

Territorial planning is the basic tool available to public decision makers to help cities evolve towards the achievement of specific objectives.

Important directions qualified and pursued in Europe during the 20th century were the *new towns* that represent extraordinary evolutions of the dynamic theories, rules and implementations. New towns mainly refer to new settlements complete with residences and services.

In the last three decades of the 20th century, a central direction of development was the *recovery of historic centers*. This direction evolved towards *urban regeneration*, which concerns built environments. In the first decades of the 21st century, urban regeneration has focused on the recovery and regeneration of areas not endowed with a specific historical interest.

In the 21st century, one of the most important directions is the *smart city*, which has the perspective of pursuing the sustainability objectives recalled in the Introduction Section of the paper.

Specific links between new towns, urban regeneration and smart cities have been proposed for territorial plans [22].

In the smart city European approach, the convergence among theories, rules and implementations considers the challenges related to the following sectors [23]:

- Transport, including supply (infrastructures and services), demand (people and freight) and their interactions;
- Information and Communication Technologies (ICT), including infrastructures, immaterial services and information and data produced for historical and real-time configuration of the system;
- Energy, including infrastructures and services for the production and distribution and consumptions.

City planning includes decisions about the configuration of infrastructures and services at different dimensions (see Figure 3) [24]:

- The territorial dimension, in relation to the spatial extension of the area affected by the decisions (e.g., local, regional or national, varying the extension from a limited to wide area);
- The temporal dimension, in relation to the temporal evolution of the potential effects produced by the decisions (e.g., strategic, tactic or operative, varying from long-term to short-term temporal evolutions);

- The deepening dimension, in relation to the level of aggregation in the analyses for measuring potential effects produced by the decisions (e.g., master, sectorial or feasibility, varying from the maximum to minimum disaggregation level of detail).

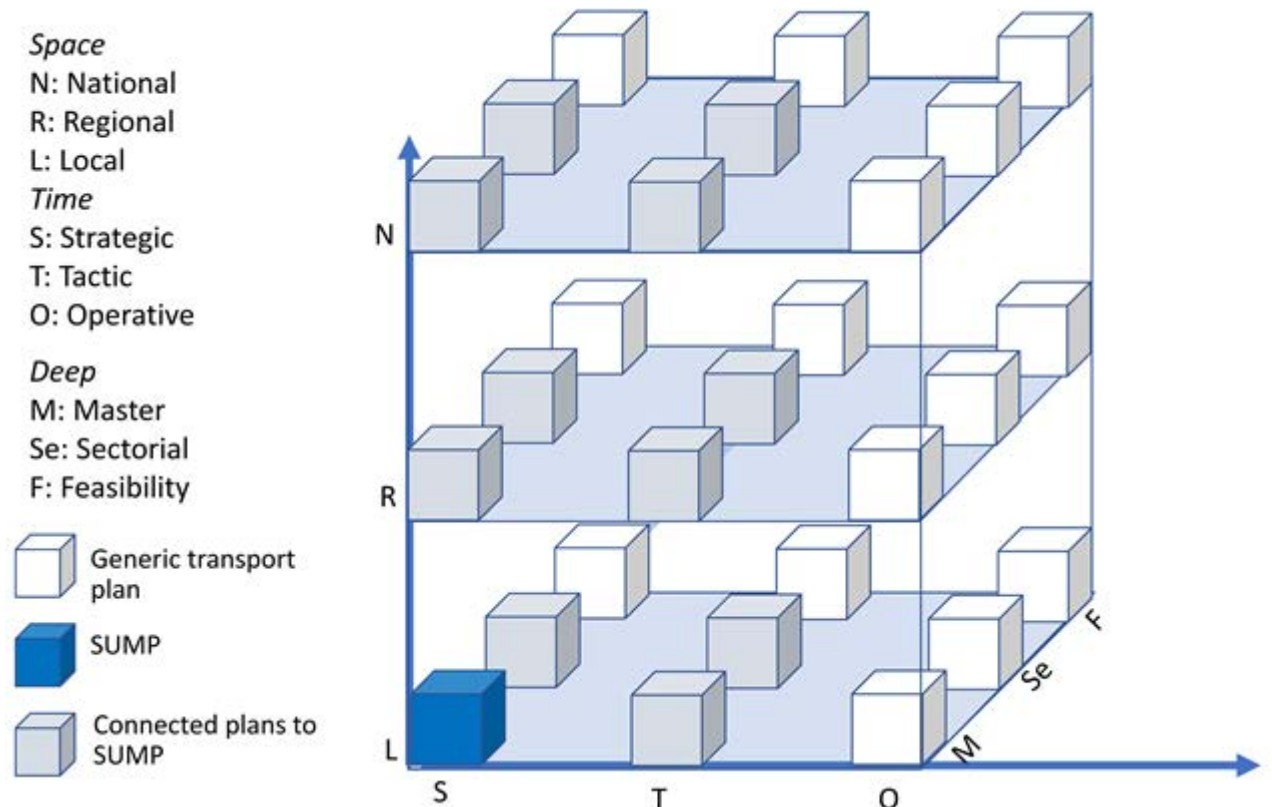

**Figure 3.** Transport planning dimensions.

The identification of these dimensions aids to comprehend the mutual interactions among different transport planning tools produced by different decision makers [25].

In the defined context, it is useful to analyze the processes that are currently in progress at two levels: at a level more identifiable as a master plan and at another as a sectorial or implementing plan. At the first level, the main attention is given to the process of the general definition of a plan for the smart city, while at the second level, the reference is the process of definition for the passenger mobility sector. The first level is explored in Section 3, while the second level is dealt with in Section 4.

### 3. The European Smart City Applied Process at the Master Level

The method for analyzing the smart city processes structured in institutional channels is divided into two levels, each of which is divided into two steps. The European development planning process of the smart city must be explored in the two levels described in Section 2 above: master and sectorial.

For the master level, the EU defines a strategic plan for mobility called the SUMP. The method used to verify the implementation of the smart city is divided into two steps. In the first step, the founding elements of the smart city are defined as presented in some European documents. In the second step, it is analyzed how these elements must be developed within the SUMPs. At the end of the second step, it is verified in how many reference cities a SUMP is implemented at the European level for smart city.

The second level investigates the urban mobility sectorial plan. In the first step, the MaaS is analyzed in its articulations, specifying the successive generations of MaaS. In the second step, it is examined how the indicators relating to the sustainability objectives

can be derived from the MaaS. Additionally, for the MaaS level, it is verified in how many reference cities a MaaS is implemented at the European level for smart city projects. Finally, the proposed complete process is verified by examining how many smart cities have both master (SUMP) and sectorial (MaaS) plans developed. For clarity, the entire survey is presented in an integrated manner.

The first level is presented in this section, and the second level is presented in Section 4.

### 3.1. The Smart City Direction of Development

The transport, ICT and energy sectors constitute the European smart city pillars according to the directives introduced in 2012 (Figure 4). Following the European indications, in the first instance, each of the three sectors is a founding element of the smart city, and none of them can be missing; in the second instance, there must be ever-increasing integration between the three sectors. For making the smart city's approach operative, the European Commission launched the European Innovation Partnership for Smart Cities and Communities (EIP-SCC). The initiative was strictly connected to European flagship initiatives for innovation in the context of European Union 2020 strategy for smart growth [24]. The main objective of EIP-SCC creation was to realize a common framework for minimizing market fragmentation, optimize efficiencies and encourage collaborations among different stakeholders across Europe [26]. Following this method, innovative solutions are studied, regulated and implemented in order to contribute to the achievement of the SDGs in European cities. In particular, the partnership contributes for reducing energy and resource consumption and greenhouse gas (GHG), at the same time improving the quality of life of citizens.

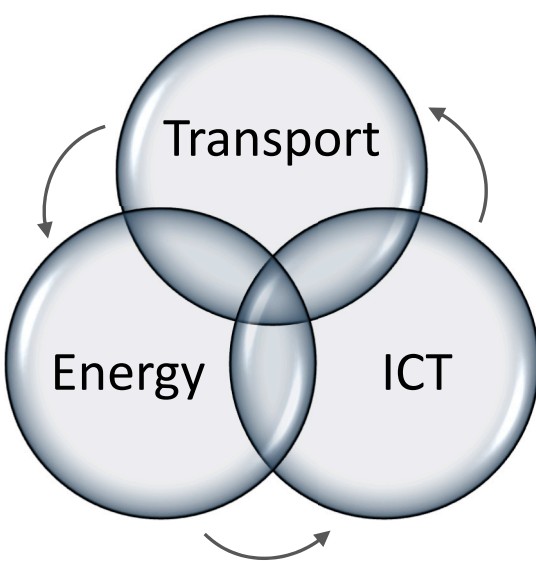

**Figure 4.** The smart city European pillars.

In the years 2013 and 2014, the EIP-SCC produced two guidelines for regulating and addressing the European smart city process:

- The Strategic Implementation Plan (SIP) [27], which indicates the priority area and domains of a smart city;
- The Operational Implementation Plan (OIP) [28], which specifies the SIP with actions and measures.

The two documents constitute the basis of EIP-SCC initiatives, including the calls under Horizon 2020. The main operative tool of EIP-SCC is the "Marketplace of the European Innovation Partnership on Smart Cities and Communities" (M-EIP-SCC), a platform that collects information about the main policies and smart city implementations at European level. The last advancement merges M-EIP-SCC and the "Smart Cities Information System"

(SCIS) into one single platform, the Smart Cities Marketplace (SCM) [29]. One of the main products of the SCM is a database that collects information about the EU-funded Smart Cities Initiatives (SCIS database) [30]. The SCM platform is also a repository of different kind of documents that can be grouped into the following categories:

- "Policy Papers" that identify the main challenges starting from a state-of-the-art process for formulating recommendations for future implementation; these documents are useful for policy makers and stakeholders involved in smart city projects;
- "SCIS Essential Monitoring Guides" that provide the preparation for the "Self-Reporting Tool", collecting key performance indicators (KPIs) measuring the results of EU SCC projects;
- "CONCERTO publications archive" that collects information about the European Research Framework Programs FP6 and FP7; the H2020 project is archived in the SCIS database.

Smart city solutions are complex urban interventions involving stakeholders with different interests and roles. It requires the "multi-aspectual" integration for defining a long-term vision, or a road map useful to overcome the barriers. Private companies and public authorities often follow short-term cycles for deciding their investments. The smart city approach requires a behavioral change for adapting their choices to emerging challenges.

According to the three processes introduced in Section 2, the theory is developed in the research centers and published in scientific journal, and the rules and the implementations are in the SCM, where information is collected. Figure 5, which is derived and elaborated from a European Report [29], shows a diagram that allows us to identify the overall state-of-the-art process of smart city developments with regards to rules and implementations; specifically, indications are given to deepen the elements relating to the 85 projects implemented and to all the rules issued at the European level:

Rules or policy recommendations, based on lessons learned from projects results, and useful for policy makers at the local, national and EU levels [31]; rules or guidelines named "Shape" for supporting smart cities' project development with "solution booklets" that illustrate technologies and concepts by means of real examples in European cities; "policy papers", which address future implementations; "recommendations on EU R&I and regulatory policies", which individuate common mechanisms that could be replicated in other cities; the "Smart City Guidance Package" or guidelines for implementing a smart city at the European level; "Action Clusters", which assemble a set of specific partners for facing specific issues related to smart cities (a specific cluster considers "Sustainable Urban Mobility");

Implementations or best practices [32], based on replicable innovations for city planners and policy developers [33]; reports on the implementations are grouped into SCM sections as follows:

"Explore", which illustrates the 85 European projects implemented in small, medium-sized towns and metropolises areas; here, there are reports on European projects about energy, ICT and urban mobility and information about 185 demonstration sites from the 85 projects;

"Deal", which describes all European initiatives aimed at creating matchmaking between cities and business; the final goal is to increase the collaboration and individuate a network of investors interested in supporting the implementation of smart city solutions.

As far as the rules are concerned, there are guidelines with issues and solutions specific for urban transport planning, for example, dedicated to stakeholder engagement, electrification of mobility and urban freight transport. For instance, "The Smart Guidance Package guidelines" reports indications that implement the "Plan-Do-Check-Act" concept for creating a vision referring to the long-term strategy, preparing and implementing the plan, following their progress and replicating the best practices in other urban contexts. As far as urban mobility is concerned, the guidelines for SUMP are the main reference for urban planning [34].

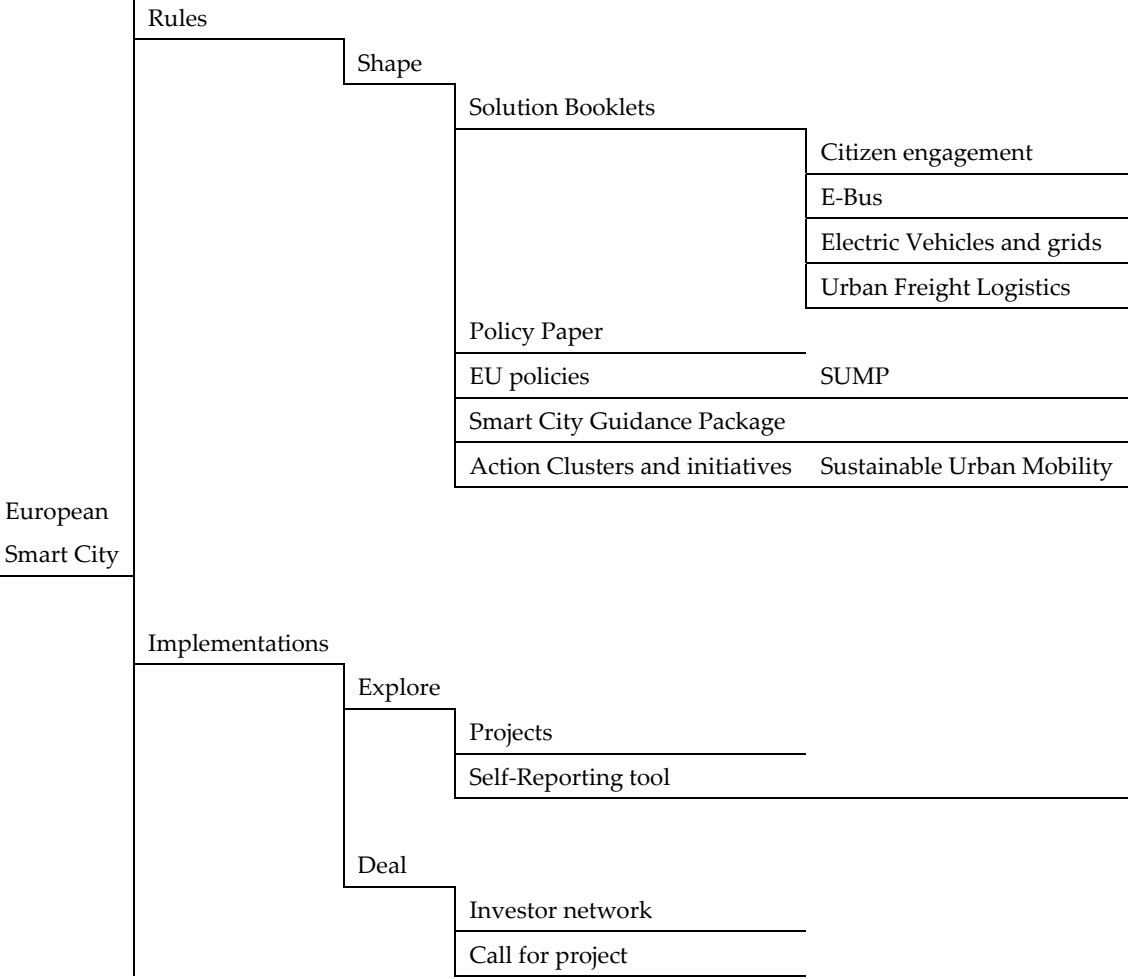

**Figure 5.** The European marketplace for smart city (elaboration from [29]).

From the analysis of the SCM platform, it is possible to identify the specific rules and implementations for the European smart city by focusing on people and freight mobility. Figure 6 recalls the main documents available in the SCM.

SIP and OIP promote the integrated planning approach for pursuing the smart city perspective. This means working without administrative boundaries but with specific temporal goals. This paper focuses on the specification of the integrated planning approach for sustainable urban mobility (Figure 6). It concerns actions for transport supply and demand that potentially contribute to urban sustainability. The concept implies an evolution of the traditional approach to transport planning. For defining planned actions, it is necessary to take into consideration the mutual interactions that the transport system has with the other civil sectors (with a priority energy and ICT). Guidelines for smart cities indicates the SUMP as the integrated planning tool that contribute to reach the convergence among transport, ICT and energy processes.

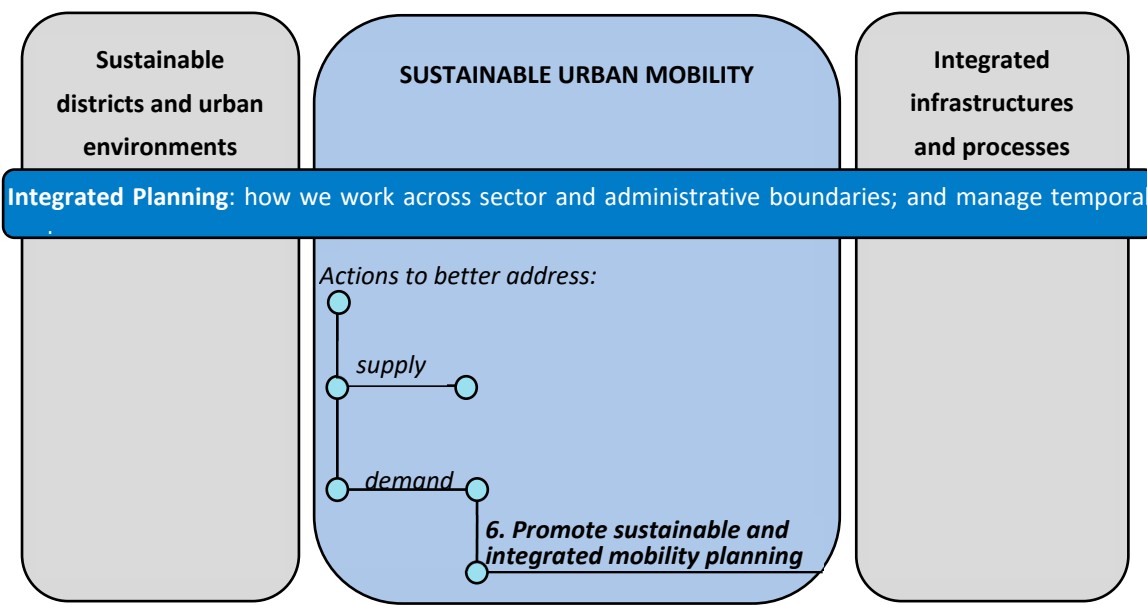

**Figure 6.** Sustainable and integrated mobility planning.

*3.2. Smart City and SUMP*

Rules refer to regulations, laws and guidelines for facing the challenges of urban mobility planning. It is necessary to clarify the different kinds of rules in relation to their role in the planning process. The SUMP is a strategic urban planning tool. According to the European indications, the SUMP can make the European smart city approach concrete.

One of the main products of the transport planning tools are the guidelines for facing urban mobility challenges. According to Russo and Rindone [24], the rules can be classified into the following classes:

- *Prescriptive* rules, for defining the boundaries within which the urban transport systems must move (e.g., financial budgets for realizing transport infrastructures); at the European level, prescriptive rules comprise regulations and directives that represent constraints (e.g., TEN-T regulations);
- *Address* rules, for promoting best practices for realizing sustainable urban transport systems; at the European level, address rules comprise guidelines for promoting objectives and strategies that represent recommendations (e.g., green papers, white papers, communications); the SUMP guidelines belong to this class [34].

The SUMP planning tool indicates the objectives and actions for implementing the MaaS paradigm, which contributes to reaching the convergence among energy, transport and ICT.

In the transportation planning process, the Sustainable Urban Mobility Plan has the following dimensions:

- The local territorial dimension, because the decisions concern the transport system of a single city or set of nearby cities;
- The strategic temporal dimension, because the decisions concern the transport system from a long-term perspective;
- The master deepening dimension, with decisions that evolve from the entire system to the feasibility of a single infrastructure.

The SUMP interacts with others planning documents at the local (e.g., territorial plans), regional (e.g., Transport Regional Plan) and national (e.g., National transport plan) levels. The SUMP contents have to face the challenges launched by SIP and OIP for smart cities. With specific references to urban mobility, the principal issues of the SUMP are:

- City logistics, for urban freight mobility;
- Door-to-door mobility for urban people mobility;

- Electrification of all forms of mobility.

The common aim is to move from a traditional concept of mobility based on physical infrastructures and a limited amount of information to a concept of smart mobility in which users' needs are at the center of the planning problems. People and freight mobility needs have to be satisfied through the combination available: ICT and energy technologies. In particular, smart urban mobility includes the following applied process regarding the different demand components (Figure 7):

- People mobility and Mobility as a Service (MaaS) concept [35];
- Freight mobility and Logistics as a Service (LaaS) [36,37], Freight as a Service (FaaS) [38] and Self-Organizing Logistics (SoL) [39,40] concepts.

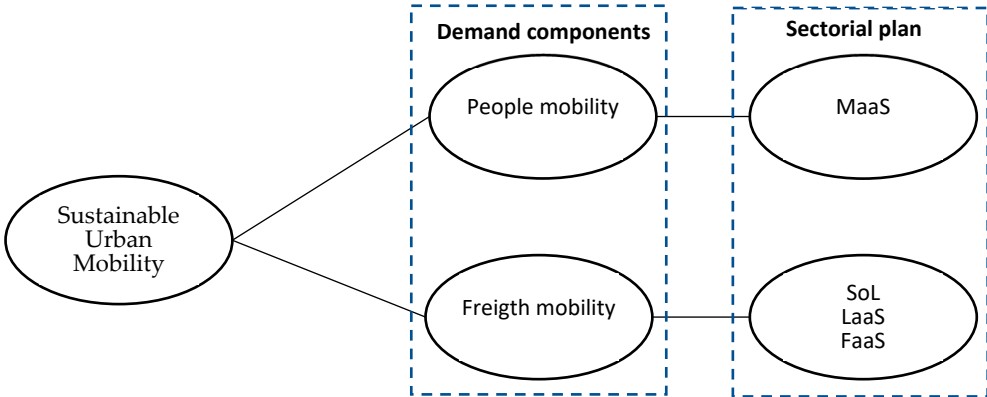

**Figure 7.** Sustainable urban mobility components.

European cities started the implementation process of the SUMPs. The European observatory ELTIS [41] monitors the level of implementations of European rules on urban mobility [42]. The results of the monitoring process are collected on the "Eltis city database", the repository of SUMPs and transport plans adopted in European cities. For each city, it is possible to verify the status of implementation of their urban mobility plans [41]. The uniformity regards the quantity and quality of the SUMP. For instance, in Italy, only 50% of metropolitan cities have adopted a SUMP [11].

An initial analysis was carried out to verify whether the path investigated, which sees the smart city as the main direction of progress for pursuing sustainability through transport plans on different time scales, is valid.

In the first step, the projects presented in the SCM platform, referred to above, and the data per city presented in the ELTIS platform concerning the development of SUMPs in cities were analyzed. Among the 85 projects presented in the SCM, 12 are dedicated to the Mobility and Transport theme. These projects involve 28 lighthouse cities, out of a total of 48, and 35 fellow cities, out of a total of 72. That is, most of the lighthouse cities for the smart city are developing a mobility project. On the other hand, the ELTIS platform was investigated by researching how many lighthouse cities (of SCM) also have a SUMP. From the data crossing it emerged that out of 28 lighthouse cities, 18 have a SUMP. It should be noted that for each city considered with a SUMP in the ELTIS database, institutional sites were checked for a link to the SUMP.

In the second step, the theme of the MaaS sectoral plan was investigated, which is analyzed in the next section, while here, for continuity of reflection, the results of the smart city–MaaS intersection are reported. For smart cities, the projects present in the SCM platform, referred to above, were investigated again, while European data were not available for MaaS projects given their recent introduction. In Italy, for example, the first three MaaS projects throughout the country are still being developed. Then, this work proceeded with a web search on the main websites of the 28 lighthouse cities, and

it emerged that 20 have a MaaS project in progress. The result in this case is even more significant because, as seen, there are only a few cities with MaaS in Europe.

Finally, a double intersection was conducted by verifying how many smart cities have SUMP and MaaS models at the same time. It emerged that 14 of the 28 smart cities have both the master plan and the sectoral plan. This allowed us to support the hypothesis of a common institutional path that sees the smart city as an important direction for both SUMP and MaaS models, since these are the two main tools of the plan to pursue sustainable development.

The European Commission (EC) is working on promoting the SUMP implementation. The new "European urban mobility framework", part of the wider "Efficient and Green Mobility Package", underlines that cities play a relevant role in the Trans-European-Transport Networks (TEN-T) [43]. In fact, the revised TEN-T guidelines consider urban mobility as essential for improving network performances, because it represents the "first and last mile" for passengers and freight mobility. The new guidelines indicate that the "largest 424 EU cities on the TEN-T network should adopt a sustainable urban mobility plan (SUMP) by 2025". For this reason, the EC announced the introduction of an upgraded SUMP concept that has clear priorities "to favour sustainable solutions including active, collective and public transport and shared mobility". In this context, MaaS models represent a solution for supporting public transport and increasing sustainability.

## 4. The European Smart City Applied Process at the Sectorial Level

The territorial smart city process in the urban area at the sectorial level of people mobility has assumed the form called Mobility as a Service (MaaS) in the last few years.

The MaaS paradigm implies the necessity to improve connections among ICT, transport and energy to achieve the urban mobility sustainability goals. Then, it is necessary to clarify the advancements on transport system theories in relation to potentialities offered by ICT and energy and the achievable outputs, which are useful for estimating externalities produced by planned configurations of urban transport systems.

### 4.1. The MaaS Levels

The Maas paradigm may be a better solution for facing urban mobility challenges. The paradigm is enabled by emerging ICT, which allow transport operators to design, manage and realize mobility services tailored on specific travel user needs [44] using advanced energy vehicles. The ICT support operators manage information flows for historical configurations in real time. However, as with other smart solutions, MaaS requires a new state among theories, rules and implementations. In the following, transport system theories are focused on in relation to implications deriving from emerging ICT.

By selecting the applications in transport systems at the urban level, the emerging ICT actually used in the city are as follows ([13,37,45]):

- Internet of Things (IoT) for exchanging of data between objects;
- Big data (BD) for managing large amounts of data in real time, in terms of volumes, variety and velocity;
- Blockchain (BC) for managing transactions in value or sensitive data between decision makers;
- Artificial intelligence (AI) for making decisions between sets of alternatives.

It is necessary to also recall digital twin, which can be used for virtual representation of a real-world physical system. The recalled emerging ICT are used in some transport sectors but without a systematic approach for implementing the MaaS paradigm.

For making the MaaS solutions effective, it is essential to improve and enhance knowledge of the mobility phenomenon. Some authors study travelers' attitudes towards MaaS [46], but urban complexity requires improvement on theories about the transport system and their components: the supply subsystem, including transport infrastructures and services [4]; the demand subsystem, including people and business mobility needs and their travel choices [17]; and demand–supply interactions, for reproducing how people

and business mobility needs are satisfied by infrastructures and services [16]. Classical Transport System Models (TSM) can be enhanced by integrating them with emerging ICT and specific energy cycles. To evolve MaaS, it is necessary to define theoretical advances that explicitly insert the emerging ICT in the TSM.

ICT and TSM became parts of Decision Support Systems (DSS), which support public authorities, companies for transport system design and management. DSS enable a priori and a posteriori quantitative evaluations of sustainability goals and their achievements measured by distances from targets [13].

Recently, in the literature [16], different levels of advancements for the MaaS paradigm have been introduced (Figure 8):

- N-MaaS (No MaaS), comprising single and separate transport systems without integration;
- MaaS 1.0 or I-MaaS, comprising a transport system with services integrated by means of an ICT platform used by operators and users (ICT MaaS); in this level, different sublevels can be identified in relation to the typology of provided services (e.g., information or financial transactions);
- MaaS 2.0 or T-MaaS, comprising I-MaaS systems enhanced by TSM that support transport system designing and management inside a DSS platform; with emerging ICT and theoretical advancements of classical TSM, it is possible to better understand the mobility phenomena;
- MaaS 3.0 or S-MaaS (Sustainable, TSM and ICT MaaS), comprising I-MaaS systems and T-MaaS enhanced transport system design and management, considering the sustainability goals and targets, recalled in the Introduction Section, by means of Space Economic Transport Interaction (SETI) models and Environmental Impact Functions (EIFs).

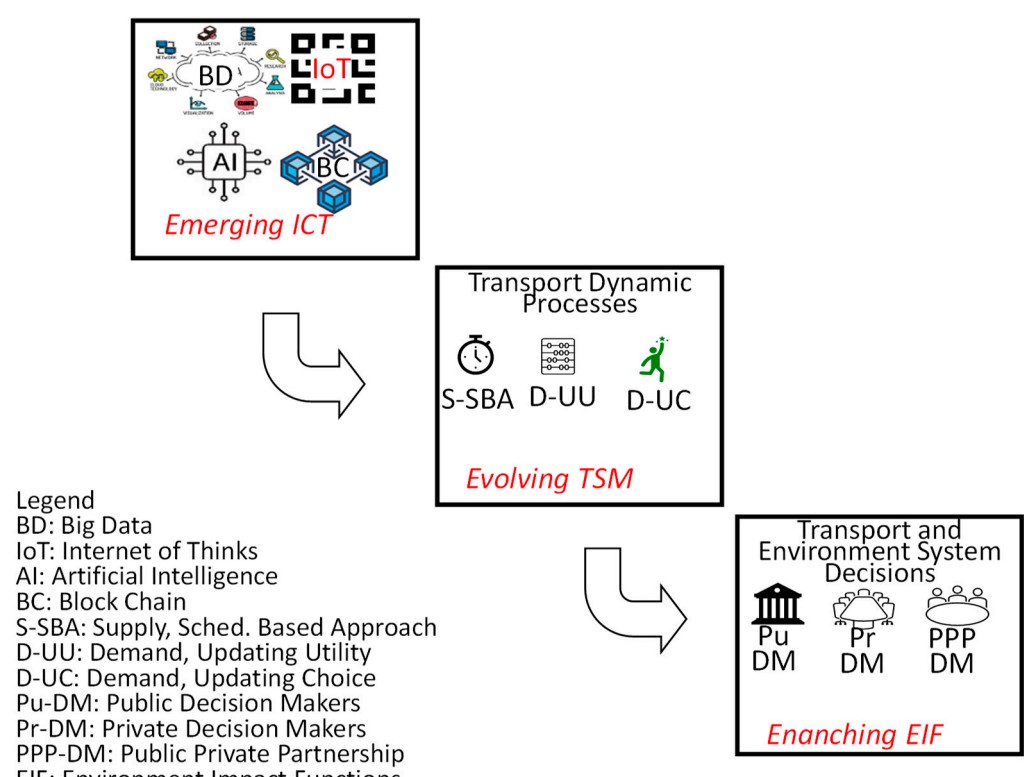

**Figure 8.** Advancements of the MaaS paradigm.

Emerging ICT (artificial intelligence, big data, internet of things, block chain, digital twin, etc.) work for the development of DSS together with TSMs and EIF for supporting

decision makers to individuate system configurations based on the mobility needs pursued by the SDGs.

Different theoretical issues, regarding single components and their interactions within a unique MaaS platform remain open. Among the final aims of the European smart city approach, the MaaS paradigm has to ensure not only the convergence between transport and ICT, but also convergence with energy issues for achieving sustainability goals recalled in the Introduction Section of this paper.

*4.2. Transport System Models (TSM) and Environmental Impact Functions*

As synthetically presented in Section 4.1, emerging ICTs enrich the modelling capabilities of TSM. They expand information and data about transport systems and their sub-components. This has theoretical implications on TSM, in particular for modelling transport supply and demand.

In relation to transport supply modelling, it is necessary to evolve the classical approaches that represent the system from a static condition (i.e., without explicitly considering the temporal evolution of performances) to a dynamic approach by considering the temporal lags (day-to-day t and within day τ) introduced by Russo [13]. The MaaS paradigm imposes the adoption of a schedule-based approach to represent the temporal connections inside the same transport mode or modes supplied by different operators (e.g., urban and extra-urban buses), as well as to forecast travel times.

In relation to demand modelling, information and data driving from emerging ICT make the evolutions of classical approaches towards dynamic processes possible. The dynamic approach is needed for considering that ICT expand the user's knowledge on the whole transport network in the previous days before the day t at the time τ. This implies the specification of two models:

- A utility update model for considering the different users' perceptions of travel options and the relative utilities, in relation to current and previous knowledge and the information derived from ICT;
- A choice update model for considering the influence of information on the travel choice mechanism in a generic instant (τ) of the generic day (t).

In the following, according to the general formulations reported in [13], there is the specification of updated utilities, with the availability of big data (BD) and Internet of Things (IoT) technologies.

The utilities of the generic path k at the generic day t and instant τ, $v_k[t, \tau]$, can be expressed with the following formulation:

$$v_k[t, \tau] = v(v_k^{BD} [t - 1, \tau], g_k^{IoT} [t, \tau - 1]) \tag{1}$$

where $v_k^{BD} [t - 1, \tau]$ is the travel utility at the day $t - 1$, calculated and stored with the support of BD; $g_k^{IoT} [t, \tau - 1]$ is k path cost experienced at the time $\tau - 1$ of day t, supported by IoT.

The formulations referred to apply, recursively, to all days with similar characteristics, such as all working days. In this way, the generic day (t) can be any working day of any week.

TSM provide simulations of the transport system in relation to current and planned configurations defined for achieving sustainability goals [47]. Simulations produce data and information for the ex-ante evaluation of internal effects produced by decisions in the transport system. However, the verification of sustainability goals requires researcher to obtain an estimation of the effects produced in the external environment (externalities). For supporting quantitative ex ante evaluations, the outputs produced by TSM are used in specific impact functions, or EIF. EIF relate transport flows with the specific externalities (e.g., road accidents, environmental emissions, . . . .), by using physical and functional

parameters. A specification of EIF regards each link of the transport network, and it can assume the following form:

$$\mathbf{e}^* = e(\mathbf{f}, \lambda) = e(\Delta \times \mathbf{h}, \lambda) \tag{2}$$

where:

$\mathbf{e}^*$ is the vector of the impact values;

$e$ is the vector of the impact functions;

$\mathbf{f}$ is the vector of link flows;

$\mathbf{h}$ is the vector of path flows;

$\Delta$ is the link–path incidence matrix;

$\lambda$ is the set of physical and functional parameters.

In the literature, there are works reporting the calibration of the $\lambda$ parameters ([48]). Note that TSM play a relevant role in the estimation of the externalities produced by the transport system [49], considering also the walk accessibility [50].

## 5. Conclusions

The smart city concept has implications in the theories, rules and implementation of the planning process of territorial systems. The paper focused on the European approach to smart cities. The European Commission has individuated transport, energy and ICT as the three sectors through which processes determine the urban state of a smart city.

The paper focused on the two levels of planning, the master and sectorial levels, by means of which the smart city direction is applied. Specific attention is on the European rules that comprise the law, directives and guidelines. The master planning level is the SUMP, which asks to city administrators to adopt a transport system approach for achieving urban sustainability in relation to people and freight mobility. The main paradigm promoted by European SUMP guidelines for people's mobility at the sectorial level is the MaaS paradigm. By means of the MaaS paradigm, it is possible to integrate the smart city European pillars: transport, energy and ICT. The introduction of information derived or processed by emerging technologies implies new specifications and the calibration of the classical transport user's behavioral models.

The implementation of the MaaS paradigm has further implications that constitute the objectives of further developments of this research: *theories* about advancements on the territorial planning approach and related modelling frameworks—for instance, spatial economic and transport interaction models for simulating systems at the medium and long term—and the calibration of models for updating choices (mode, services and paths); *rules* about the kind of institutions and governance models for regulating, planning and producing MaaS solutions (e.g., market-led or government-led models); *implementations* about advancements on material and immaterial infrastructures, including technologies, big data and IoT applications. To test the effectiveness of the proposed route, the reference European cities for smart city projects were analyzed, and the results showed that most of them have a SUMP or MaaS and that half have both plans, confirming the process identified.

The research carried out provides an important contribution because it allows us to identify a homogeneous path, in urban areas, from the goals of Agenda 2030 to the daily implications for citizens, allowing technicians and researchers to correctly define their research within this homogeneous path. The limits of the research concern the study being carried out only for the reality of the European Union; it would be useful to develop a similar study for other large geographical areas, such as Asia, America, Oceania and Africa. In each of the continents, there are particularly important territorial development dynamics.

The obtained results are useful for different actors. Researchers can benefit from the results because they can work on theories' advancements; planners and decision makers can increase their knowledge about European opportunities for urban sustainability and its implementation.

**Author Contributions:** Conceptualization, F.R. and C.R.; methodology, F.R.; validation, C.R.; formal analysis, F.R.; investigation, C.R.; writing—original draft preparation, C.R.; writing—review and editing, F.R.; visualization, C.R.; supervision, F.R.; All authors have read and agreed to the published version of the manuscript.

**Funding:** This research was partially supported by the Dipartimento di ingegneria dell'Informazione, delle Infrastrutture e dell'Energia Sostenibile, Università Mediterranea di Reggio Calabria and by the project "La Mobilità per i passeggeri come Servizio—MyPasS", Fondi PON R&I 2014–2020 e FSC "Avviso per la presentazione di Progetti di Ricerca Industriale e Sviluppo Sperimentale nelle 12 aree di Specializzazione individuate dal PNR 2015–2020", codice identificativo ARS01_01100.

**Informed Consent Statement:** Not applicable.

**Data Availability Statement:** Not applicable.

**Acknowledgments:** The author would like to thank all the researchers from the Transportation Group at Department DIIES for the long and deep discussions. The author also wishes to thank the reviewers both for the careful analysis of the proposed formulations and for the strong request to anchor each crucial step in the comparison with the most advanced international scientific literature. It is evident that every error is the responsibility of the author. The author would also like to thank MDPI for supporting this paper with the 100% of the author processing charge.

**Conflicts of Interest:** The authors declare no conflict of interest.

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
