# Peer review of "Smart City for Sustainable Development: Applied Processes from SUMP to MaaS at European Level"

_applsci, doi:10.3390/app13031773_

Round 1

Reviewer 1 Report

Very good article. I just have a few minor comments.

row 39-41 no source 

row 45 - where?

row 49 no source 

fig 4. Are these pillars spinning? What does common mean?

Does formula 1 apply for every day of the week? for example t = Monday.

Poor knowledge of the work of other scientists. The authors quote themselves as many as 18 times!

Author Response

Dear Reviewer 1, We would first express to thank for the work done, with the special check of all the formulation, and the suggestions proposed, which We introduce in the new version and the details of which are reported below.

Reviewer 2 Report

It examines the development of European smart cities by concentrating on  urban mobility issues and how they relate to the other two pillars. The main concern of this study is lacking to figures and graphs. Also, the novelty is another concern. I think authors must be able to justify the novelty of their research in a separate section like 'goals and objectives'. There you should also describe the research limitations and contribution to the body of the literature.

The method used to analyze the smart city applied process
at European level inside the regulation framework is not specified well.

authors are suggested to separate the conclusions from discussion.

Author Response

Dear Reviewer 2, We would first express to thank for the work done, with the special check of all the formulation, and the suggestions proposed, which We introduce in the new version and the details of which are reported below.

Reviewer 3 Report

Abstract lacked for presenting the objective, contribution and methodology.

Results are not clear.

References should be numbered according to their appearance order. For references appearance in the text, it is a strange to mention the author, date and a number. Please standardize.

Lines 230 to 232 are your own findings? or belong to others? if so, please provide the reference. 

Author Response

Dear Reviewer 3, We would first express to thank for the work done, with the special check of all the formulation, and the suggestions proposed, which We introduce in the new version and the details of which are reported below.

Round 2

Reviewer 2 Report

The authors answered some comments and ignored others, also, in the revised version, there is no highlight for the newly inserted sentences. I believe the manuscript still need major corrections for identifying the novelty of the study and what the new contribution can this study add to science. In addition, the study lacks to results and figures. Furthermore, the methods of analysing the European smart city process need to explain more.

Author Response

Dear Reviewer 2, We would express again to thank for the new work done and the suggestions proposed.

In the rest of this letter We report the changes and additions You requested and the modification introduced in the new version.

The new changes made after the second review (by reviewer 2) are shown below. The second part recalls all the changes made after the first review (by reviewer 2).

Reviewer 3 Report

Still need more research effort. Current form looks like a review paper.

Author Response

Dear Reviewer 3, We would express again to thank for the new work done and the suggestions proposed.

In the rest of this letter We report the changes and additions You requested and the modification introduced in the new version.

The new changes made after the second review (by reviewer 3) are shown below. The second part recalls all the changes made after the first review (by reviewer 3).
